# Porcine Reproductive and Respiratory Syndrome Virus Interferes with Swine Influenza A Virus Infection of Epithelial Cells

**DOI:** 10.3390/vaccines8030508

**Published:** 2020-09-05

**Authors:** Georges Saade, Déborah Ménard, Caroline Hervet, Patricia Renson, Erika Hue, Jianzhong Zhu, Laurence Dubreil, Romain Paillot, Stéphane Pronost, Olivier Bourry, Gaëlle Simon, Joëlle Dupont, Nicolas Bertho, François Meurens

**Affiliations:** 1BIOEPAR, INRAE, Oniris, 44307 Nantes, France; georges.saade@inrae.fr (G.S.); deborah.menard@oniris-nantes.fr (D.M.); caroline.hervet@inrae.fr (C.H.); nicolas.bertho@inrae.fr (N.B.); 2ANSES, Ploufragan-Plouzané-Niort Laboratory, Swine Virology Immunology Unit, 22440 Ploufragan, France; patricia.renson@anses.fr (P.R.); olivier.bourry@anses.fr (O.B.); gaelle.simon@anses.fr (G.S.); 3LABÉO Frank Duncombe, 14280 Saint-Contest, France; erika.hue@laboratoire-labeo.fr (E.H.); romain.paillot@laboratoire-labeo.fr (R.P.); stephane.pronost@laboratoire-labeo.fr (S.P.); 4Normandy University, UNICAEN ImpedanCELL, Federative Structure 4206 ICORE, 14280 Saint-Contest, France; 5College of Veterinary Medicine, Comparative Medicine Research Institute, Yangzhou University, Yangzhou 225009, China; jzzhu@yzu.edu.cn; 6Joint International Research Laboratory of Agriculture and Agri-Product Safety, Yangzhou 225009, China; 7PAnTher, INRAE, Oniris, 44307 Nantes, France; laurence.dubreil@oniris-nantes.fr; 8PRC, INRAE, 37380 Nouzilly, France; joelle.dupont@inrae.fr

**Keywords:** pig, PRRSV, swIAV, coinfection, epithelial cell, interference

## Abstract

Respiratory infections are still a major concern in pigs. Amongst the involved viruses, the porcine reproductive and respiratory syndrome virus (PRRSV) and the swine influenza type A virus (swIAV) have a major impact. These viruses frequently encounter and dual infections are reported. We analyzed here the molecular interactions between viruses and porcine tracheal epithelial cells as well as lung tissue. PRRSV-1 species do not infect porcine respiratory epithelial cells. However, PRRSV-1, when inoculated simultaneously or shortly before swIAV, was able to inhibit swIAV H1N2 infection, modulate the interferon response and alter signaling protein phosphorylations (ERK, AKT, AMPK, and JAK2), in our conditions. SwIAV inhibition was also observed, although at a lower level, by inactivated PRRSV-1, whereas acid wash treatment inactivating non-penetrated viruses suppressed the interference effect. PRRSV-1 and swIAV may interact at several stages, before their attachment to the cells, when they attach to their receptors, and later on. In conclusion, we showed for the first time that PRRSV can alter the relation between swIAV and its main target cells, opening the doors to further studies on the interplay between viruses. Consequences of these peculiar interactions on viral infections and vaccinations using modified live vaccines require further investigations.

## 1. Introduction

Respiratory infectious diseases are still a major concern in pigs [1]. Most often multiple infectious agents including bacteria, parasites, and viruses are involved in the development of clinical respiratory conditions [2]. Thus, the reductionist approach considering only single infections might miss important interactions [2]. The term “coinfection” describes simultaneous infections of the host with different microorganisms while “superinfection” refers to a succession of infections where one microorganism infects the cell or the host before any infection by a second “superinfecting” microorganism [2,3]. Amongst viruses causing respiratory coinfections and superinfections in pigs, the porcine reproductive and respiratory syndrome virus (PRRSV) and the swine influenza type A virus (swIAV) are of major importance because of their economic impacts and their prevalence worldwide [2]. Indeed, epidemiologic studies in France, Korea, and the USA have observed associations between these two viruses in pigs [4,5,6,7,8]. Both viruses are RNA viruses, PRRSV targeting mostly alveolar macrophages, and swIAV infecting predominantly epithelial cells in the upper and the lower respiratory tracts [9,10]. Two different species, PRRSV-1 (alternatively Betaarterivirus suid 1), from European origin, and PRRSV-2 (alternatively Betaarterivirus suid 2) from North American origin, are described for PRRSV [11]. The three main IAV subtypes encountered in pigs are H1N1, H1N2, and H3N2, however many genetic lineages and antigenic variants within these subtypes are co-circulating [10]. To control PRRSV infections, modified live viruses (MLV) are used as vaccines [9] and both wild type and vaccine strains persist for months in the animal, increasing the risk of dual PRRSV/swIAV infections [9]. In the literature, the clinical outcomes of PRRSV/swIAV coinfections and superinfections varied depending on the experimental settings and the studied viral strains [12,13], and the molecular consequences are still poorly evaluated and understood [2]. In previous studies, we first observed that swIAV H1N1 interfered with PRRSV-2 productive infection of alveolar macrophages (AM) [14]. We secondly demonstrated that PRRSV-2 interfered with swIAV H1N1 infection of genetically modified newborn pig trachea epithelial cells (NPTr) expressing CD163, the main receptor of PRRSV [15]. This interference using NPTr CD163^+^ cells was not surprising since, in that situation, PRRSV-2 was able to infect the same host cell as swIAV. However, this study did not address the question of interference in physiological conditions of non-modified respiratory epithelial cells.

In the current study, we established protocols to further deciphering viral interactions between PRRSV and swIAV and analyzed the generated data in the context of the available knowledge in the field [2,16]. Viral interference and its impact on both interferon response and induced signaling pathways were assessed in NPTr cells and/or lung slices using local PRRSV-1 and H1N2 swIAV strains that are co-circulating in pigs as previously shown [8]. 

## 2. Materials and Methods 

### 2.1. Newborn Pig Tracheal Epithelial Cell Line

NPTr cells [17] were cultured with Dulbecco’s modified Eagle medium (DMEM) (Eurobio scientific, Les Ulis, France) supplemented with 10% fetal calf serum (FCS) (Eurobio scientific) and 1% of Streptomycin/Penicillin/Amphotericin (SPA) solution (Eurobio scientific). Flasks were then incubated at 37 °C in a 5% CO_2_ humidified environment and sub-passages were carried out once a week to assure the cell line maintenance. 

### 2.2. Alveolar Macrophages

Alveolar macrophages used for the viral titration were obtained from bronchoalveolar lavage (BAL) of lungs collected from 5 to 7-month-old Large White conventionally bred sows. The animals were bred in accordance with European regulations by the experimental unit of animal physiology of Orfasière (UE PAO), Nouzilly, France. They were serologically tested frequently and known to be free of any common viral infection (swIAV, PRRSV, Porcine Circovirus 2, amongst others). In order to reduce the use of animals, the lungs were collected from pigs slaughtered in the course of the regular management of the experimental unit’s herds. As a consequence, no trial number has been attributed since an experimental authorisation was not requested. Once isolated, the lung airways were infiltrated with 250 mL of phosphate buffered saline (PBS) (Eurobio scientific) supplemented with 2 mM EDTA (Sigma-Aldrich, Saint-Quentin, France). The BAL was then collected, centrifuged, and passed through 40 µm cell strainers. After treatment with erythrocyte lysis buffer (10 mM NaHCO_3_, 155 mM NH4Cl, and 10 mM EDTA), AM were washed with PBS, counted and seeded onto sterile plates and flasks for virus titration and propagation. Roswell Park Memorial Institute medium (RPMI) 1640 medium (Eurobio scientific) supplemented with 10% FCS and 2% of SPA solution was used for AM culture.

### 2.3. Virus Propagation, Purification and Titration

SwIAV H1N2 strain A/Sw/Ille-et-Vilaine/0415/2011 was selected among the collection of the French National Reference Laboratory for Swine Influenza (ANSES, Ploufragan, France). It was isolated from a nasal swab taken from a pig with acute respiratory disease in a herd located in Brittany, France. It was propagated on Madin-Darby Canine Kidney (MDCK) (ATCC reference CCL-34) cells for 24 hours (h) in DMEM medium supplemented with 10% FCS, 1% of SPA solution, and 2 µg/mL of trypsin TPCK treated (Worthington Biochemical Corp., Lakewood, NJ, USA). The supernatant was then collected, clarified by centrifugation (600× *g*, 20 min) and stored at −80 °C. 

PRRSV-1 strain PRRS-FR-2005-29-24-1 (*Finistère* strain; genotype 1.1) (ANSES’ collection) was isolated in Brittany, France in 2005 from a herd with abortions. The propagation was performed on AM cultured in RPMI 1640 medium (Eurobio scientific) supplemented with 10% FCS and 2% of SPA solution for 72 h. The DV strain of PRRSV-1 (modified live virus, MLV) used in Porcilis^®^ PRRS vaccine (ANSES’ collection—MSD Animal Health) was propagated on the monkey epithelial cell line MARC-145 (ATCC reference CRL-12231). 

Aujeszky’s disease virus (ADV) strain Kojnok [18] (ANSES’ collection) was propagated on NPTr cells for 24 h in a DMEM medium supplemented with 10% FCS and 1% of SPA solution. The supernatant was then collected, clarified by centrifugation (600× *g*, 20 min) and stored at −80 °C.

All viruses were concentrated and purified on Amicon Ultra-15 centrifugal Filters (Sigma-Aldrich – reference number UFC910024 – pore size 100 kDa Nominal Molecular Weight cu-off) after a 20 min centrifugation at 4000× *g* and 4 °C. Titer determinations of swIAV H1N2 and ADV were carried out on MDCK, PRRSV-1 Finistère strain titration was done on AM, and DV strain of PRRSV-1 was performed on MARC-145 using TCID_50_ assay protocol. The viral titers of purified swIAV H1N2 and ADV reached 10^7^ TCID_50_/mL and 10^6^ TCID_50_/mL, respectively, while PRRSV-1 stock titer was 10^6^ TCID_50_/mL.

### 2.4. Real-Time Monitoring of SwIAV Cytopathic Effects Using Real-Time Cell Analysis

The xCELLigence real-time cell analysis (RTCA) is a recently developed tool allowing the assessment of cell growth and cytopathic effects using impedance in culture wells equipped with gold electrodes [19]. The presence of adherent cells in the wells alters measured impedance. This property can be used to monitor cell viability, migration, growth, spreading, proliferation, and any modification due to viral cytopathic effect [20,21]. Monitoring of real-time cell impedance was performed using the RTCA MP system (ACEA Biosciences, Montigny le Bretonneux, France) provided by LABÉO Frank Duncombe, Saint-Contest, France (https://www.impedancell.fr/). A dimensionless value, called cell index (CI) was measured and represented after derivation of impedance signals registered by the sensor analyzer of the monitor. Assessment of the background reading was carried out by filling all the wells of the E-plate VIEW PET (ACEA Biosciences) with DMEM. NPTr cells were seeded at 3 × 10^4^ cells per well, incubated for 30 min at room temperature as per the manufacturer’s recommendations, and finally transferred to the RTCA-MP station placed in the incubator at 37 °C and 5% CO_2_. After 24 h, cells were washed twice with PBS and infected with swIAV H1N2, PRRSV-1, or both viruses. The plate was placed back and CI values were measured automatically every 10 min for 72 h. A normalization of the CI values was performed using the RTCA software version 2.0 (ACEA Biosciences) to match the last point before cell infections with the viruses [19]. 

### 2.5. Virus Inactivation

After propagation of PRRSV-1 on AM, part of the viral stock was inactivated by adding beta-propiolactone (BPL) (Sigma-Aldrich) (1:4000 dilution) after incubation for 2 h at 37 °C in a 5% CO_2_ humidified atmosphere. Inactivated viral stock was clarified, purified (as described above), and stored at −80 °C. The efficiency of the inactivation process was tested by a reverse transcription-quantitative polymerase chain reaction (RT-qPCR) targeting PRRSV-1 (see PCR section) on supernatant obtained after inoculation of the AM with the inactivated virus over three passages.

### 2.6. Precision-Cut Lung Slices

Lungs were collected from 5- to 7-month-old Large White conventionally bred sows from UE PAO, Nouzilly, France. The diaphragmatic lobes were filled with a 37 °C warm mix of low-melting agarose (Sigma-Aldrich) and RPMI 1640 supplemented with 1% SPA, 10 µg/mL enrofloxacin (Bayer Animal Health, Leverkusen, Germany) and 1 µg/mL clotrimazole (Sigma–Aldrich). The lungs were transported on ice to ensure the solidification of the agarose. Transverse cuts were done and tissue was stamped out as cylindrical portions using a coring tool, then 250 µm slices were prepared in a Krumdiek tissue slicer (model MD4000-01, TSE systems, Chesterfield, MO, USA) at a cycle speed of 60 slices/min. The generated precision-cut lung slices (PCLS) were then incubated at 37 °C in a 5% CO_2_ humidified atmosphere, after adding 10% of FCS to the complete RPMI 1640 medium. The medium was changed twice with 1 h interval and twice again after 12 h. At this stage, the viability of the tissue slices was assessed by adding 10^-4^ M methacholine (acetyl-ß-methylcholine chloride, Sigma-Aldrich), and the slices were sorted and prepared for viral infections after observation of the ciliary activity under a light microscope (Olympus CKX31, Tokyo, Japan). 

### 2.7. Virus Infection and Stimulation of Newborn Pig Tracheal Epithelial Cells and Precision-Cut Lung Slices

NPTr cells were cultured in 48 well-plates for 24 h at 1–2 × 10^5^ cells per well, washed twice with PBS then infected with swIAV H1N2 at a multiplicity of infection (MOI) of 3 (enabling the infection of all the cells, according to the Poisson distribution) in the absence or presence of inactivated or live PRRSV-1 (at the same MOI). Dual administrations were carried out after variation of the order of administration and the delay between the viruses, going from 0 h to 1 h and finally 6 h (see Figure 1A, Figure 2A, Figure 3A and Figure 4A). In order to assess the importance of the virus species in the interference process and the specificity of PRRSV-1 Finistère strain effects, we also used alternatively the DV strain of PRRSV-1 used in the Porcilis PRRS vaccine (MLV in the manuscript) and ADV strain Kojnok in our dual administration protocol (Figure 1E). Then, to further decipher how PRRSV-1 can interfere with swIAV H1N2, an acid wash (40 mM citric acid, 135 mM NaCl, 10 mM KCl, pH 3.0) was applied to the cells after one-hour incubation with both PRRSV-1 and swIAV H1N2 viruses, or PRRSV-1 alone (Figure 3A). This procedure is commonly used in penetration and growth kinetic assays for enveloped viruses and is known to inactivate non-internalized viruses [22,23,24]. 

The same protocol, except acid wash, was used for PCLS infection but without any variation in the order of inoculations with 10^5^ TCID_50_/slice/virus (Figure 5A).

NPTr cells were incubated in DMEM while PCLS were kept in RPMI 1640, both supplemented with 1% of SPA solution and incubated at 37 °C and 5% CO_2_ before any harvest. Cells were harvested using a special lysis buffer from the RNeasy Mini Kit (Qiagen, Courtaboeuf, France), and PCLS were collected in Trizol reagent (Invitrogen, Cergy Pontoise, France). Supernatants were also collected and stocked at −80 °C.

### 2.8. Immune Gene Expression Analysis and Virus Detection by Quantitative Real-Time PCR

Total RNA was extracted from NPTr cells and supernatants using RNeasy Mini Kit (Qiagen) following the manufacturer’s instructions after cell lysis with RLT buffer (Qiagen). PCLS lysis in Trizol was carried out using FastPrep lysing tubes and the FastPrep homogenizer (MP Biomedicals FastPrep-24™ 5G, Illkirch-Graffenstaden, France). Then, RNA samples were treated with DNAse I Amp Grade (Invitrogen) (1 U/µg of RNA). The absence of genomic DNA contamination was validated by the use of treated RNA as a template directly in PCR. Total RNA quantity and quality were assessed using Nanophotometer (Implen, Munich, Germany). cDNA was generated with a virus reverse transcriptase in the iScript Reverse Transcription Supermix for RT-qPCR (Bio-Rad, Hercules, CA, USA) from 100–200 ng of RNA free of genomic DNA per reaction. 

The generated cDNA was then diluted (4×) and combined with primer/probe sets and IQ SYBR Green Supermix (Bio-Rad) following the manufacturer’s recommendations. To carry out the qPCR assays the selected conditions were 98 °C for 30 seconds (s), followed by 37 cycles with denaturation at 95 °C for 15 s and annealing/elongation for 30 s at optimal temperature—depending on the chosen target (Table 1). SwIAV viral transcripts, and transcripts associated with genes involved in the innate immune response such as those coding for some pattern recognition receptors, type 1 and 3 interferons, and interferon-stimulated genes were assessed. PRRSV-1 replication was not assessed here since the virus does not infect NPTr cells [25]. Most of the sequences of the primers targeting immune genes used in the study were published previously [14,26,27,28]. TLR6 and IFNλ3 specific primers were designed specifically for the current study with Clone Manager 9 (Scientific and Educational Software, Cary, NC, USA). qPCR assays were performed on a CFX96 Connect (Bio-Rad). The specificities of the qPCR assays were assessed by analyzing the melting curves of the generated products. The correlation coefficients of the standard curves were always between 0.950 and 0.995 and all the qPCR assays measured efficiencies between 90% and 110% as recommended [29]. Collected samples were normalized internally by simultaneously using the average Cycle quantification (C*q*) of three stable reference genes in each sample [30]. The three reference genes were; glyceraldehyde-3-phosphate dehydrogenase (GAPDH), beta-2-microglobulin (B2M1) and hypoxanthine phosphoribosyltransferase-1 (HPRT-1) [14,26,31]. These selected reference genes were assessed before for their stability as previously described, using geNorm [32]. Then, qPCR data (C*q*) were subjected to Genex macro analysis (Bio-Rad) [30] and expressed as relative values after Genex macro analysis.

Virus genomes, except for ADV, were amplified using Taqman qPCR assays described previously (Table 1) [33,34]. ADV specific primers were designed specifically for the current study using Clone Manager 9 (Scientific and Educational Software). The cDNA (2 µL—dilution 2×) were combined with 0.3 µL of each primer (10 µM), 0.25 µL of probe (10 µM), 5 µL of Takyon No Rox Probe MasterMix dTTP blue 2× (Eurogentec, Liège, Belgium), and ultra-pure water to reach a final volume of 10 µL. The conditions of the qPCR assays were here 95 °C for 3 min followed by 40 cycles with denaturation at 95 °C for 6 s and annealing/elongation for 15 s at 60 °C. Like the SYBR Green qPCR assays, the Taqman qPCR assays were run on the CFX96 Connect (Bio-Rad). qPCR assays efficiencies were all very close or equal to 100% as recommended in MIQE. Relative viral loads values for swIAV H1N2 in supernatants were calculated by attributing a value of 10 to the highest *Cq* score and then doubling this attributed value for every difference of 1 C*q* since the efficiency for swIAV Taqman assay was reaching 100% in our conditions. 

### 2.9. Western Blotting

NPTr cells were cultured in 48-well plates, then single or dually infected at a MOI of 3 with swIAV H1N2 and/or live or inactivated PRRSV-1. Plates were immersed in liquid nitrogen and the cells were harvested simultaneously at 5 and 10 min before being disrupted with the lysis buffer (10 mM Tris pH 7.4, 150 mM NaCl, 1 mM ethylene diamine tetraacetic acid-EDTA, 1 mM ethylene glycol tetraacetic acid, 1% (*v*/*v*) Triton X-100, 0.5% NP-40), phosphatase inhibitors (100 mM sodium fluoride, 10 mM sodium pyrophosphate, 2 mM sodium orthovanadate), and protease inhibitors (2 mM phenyl methyl sulfonyl fluoride-PMSF, 10 µg/mL leupeptin, 10 µg/mL aprotinin) (Sigma-Aldrich and Bio-Rad). Then, equal amounts of proteins were separated using sodium dodecyl sulfate polyacrylamide gel electrophoresis (SDS-PAGE) and transferred onto a nitrocellulose membrane after a 30 min incubation on ice and a centrifugation at 12,000× *g* for 20 min at 4 °C. Non-specific sites were saturated by incubating the membranes for 1 h at room temperature (RT) with Tris-buffered saline (TBS, 2 mM Tris-HCl, pH 8, 15 mM NaCl, pH 7.6), containing 0.1% Tween-20 (Bio-Rad) and 5% non-fat dry milk powder (NFDMP). Later on, membranes were incubated with the primary antibodies (working dilution 1:1000, Table 2) in TBS containing 0.1% Tween-20 and 5% NFDMP overnight at 4 °C. Then, the membranes were washed in TBS-0.1% Tween-20 and incubated for 2 h RT with a horseradish peroxidase-conjugated secondary antibody (working dilution 1:10,000). After an additional wash, proteins were detected by enhanced chemiluminescence (Western Lightning Plus-ECL, Perkin Elmer, Courtabœuf, France) using a G:Box SynGene (Ozyme, Saint-Quentin-en-Yvelines, France) coupled with the GeneSnap software (Syngene UK, Cambridge, UK, release 7.09.17). GeneTools software (Syngene UK, release 4.01.02) was selected for quantification of detected signals. Then, the results were normalized and expressed as the signal intensity in arbitrary units.

### 2.10. Immunofluorescence Analyses

Before staining, PCLS were fixed with acetone:methanol (50:50) (VWR International, Radnor, USA) while cells were fixed with 3% paraformaldehyde (Sigma-Aldrich) and permeabilized with 0.2% Triton ×-100 (Sigma-Aldrich). SwIAV H1N2 infected cells were identified after using a mouse monoclonal antibodies targeting the viral nucleoprotein (dilution 1/50) (OBT0846, clone: 1341, Bio-Rad) followed by an appropriate goat anti-mouse secondary antibody coupled to Alexafluor555 (dilution 1/100) (Ref: A21121, Invitrogen). Cells infected with PRRSV-1 were identified by using a monoclonal antibody recognizing the viral N protein (dilution 1/100) (mouse IgG1, Ref: BIO 276, Bio-X Diagnostic, Rochefort, Belgium). A goat anti-mouse secondary antibody coupled to Alexafluor488 was then used (dilution 1/100) (Ref: A21137, Invitrogen). Cy3-labeled monoclonal antibody recognizing beta-tubulin (dilution 1/100) (C4585, Clone TUB 2.1, Sigma-Aldrich) was selected as ciliated cell marker, and anti-MHC-II antibody (dilution 1/100) (clone MSA3 from monoclonal antibody center Washington State University—Pullman, WA, USA) was used as a marker for macrophages and other antigen-presenting cells. Cell nuclei were stained after incubation with 4’,6’-diamidino-2-phenylindole (DAPI) (Sigma-Aldrich) for 10 min at room temperature. The cells and the PCLS were finally washed with PBS, mounted in Mowiol 4-88 (Sigma-Aldrich). Tissue sections were covered by micro-cover glass (13 mm) (Dominique Dutscher SAS, Brumath, France). Images were generated using a ZEISS LSM 780 laser-scanning microscope (Carl Zeiss Microscopy, Jena, Germany) equipped with solid-state lasers 405, 561, and 633 nm and argon laser 488 nm.

### 2.11. Statistical Analyses

The expression of viral and cellular transcripts in cells and lung slices was expressed as relative values and data are expressed as mean ± standard deviation (SD). Due to the non-normal distribution, all data were sorted by rank before performing an ANOVA test using GraphPad Prism (GraphPad Software version 7.0, San Diego, CA, USA). Finally, Tukey’s test was performed to compare the means of the ranks among different groups. *p*-values less than 0.05 were considered statistically significant. Only significant differences between single swIAV H1N2 condition and other infection conditions as well as differences between the various coincubation conditions were indicated in Figure 1, Figure 2, Figure 3, Figure 4 and Figure 5. In Figure 6, all the significant differences are shown except those involving the control conditions.

## 3. Results

### 3.1. Simultaneous PRRSV-1 Contact with NPTr Epithelial Cells Strongly Decreases swIAV H1N2 Replication and Immune Gene Expression

To assess the impact of PRRSV-1 on infection of epithelial cells by swIAV H1N2, we first monitored the survival of NPTr cells upon inoculation of swIAV H1N2 or PRRSV-1 or both viruses simultaneously using xCELLigence Real-Time Cell Analysis (RTCA) (Figure 1A,B). In the control conditions (grey curve) the normalized cell index (CI_n_) increased over time due to cell adhesion and proliferation. In swIAV H1N2 infected culture conditions, the CI_n_ began to clearly decrease from 8 hours post-infection (hpi) (Figure 1B; pink curve), which corresponds to the induction of the virus-mediated cytopathic effects (CPE). The decrease in the CI_n_ values intensified to reach zero around 32 hpi (Figure 1B), with no more adherent cells present in the wells at 48 hpi. Conversely, at 72 hpi, in PRRSV-1 inoculated culture conditions (Figure 1B; blue curve), the CI_n_ values were not statistically different from CI_n_ values observed in control conditions, in agreement with the known incapacity of PRRSV-1 to infect epithelial respiratory cells. Interestingly, in coinoculated cell cultures, the swIAV H1N2-induced CPE was completely abolished, as the CI_n_ values remained high over 72 h (Figure 1B; red curve).

In a second experiment, swIAV H1N2 and PRRSV-1 were added simultaneously or separately to NPTr cells for 24 h before the extraction of total RNA from the infected cells. Then, the expression of swIAV viral transcripts and the expressions of genes coding for innate immune mediators were assessed (Figure 1C for the most significantly affected genes and Appendix A for all the data). 

The expression of swIAV viral transcripts—namely swIAV M-encoding gene—was significantly reduced in the coincubation condition compared to swIAV H1N2 single infection (*p* < 0.01). Thus, the addition of PRRSV-1 in culture medium resulted in a reduction of swIAV H1N2 replication. These results were further confirmed after titration of the virus in the supernatant of the different conditions using TCID_50_ assay (data not shown). Similarl to what we observed with swIAV viral transcripts, the gene expression of innate immune mediators was significantly reduced in the coincubation condition compared to swIAV H1N2 infection (*p* < 0.01) (Figure 1C and Appendix A). Microscopic analysis of cells after 24 h infection confirmed the massive infection of NPTr cells by swIAV H1N2 virus alone. However, coincubation of swIAV H1N2 with PRRSV-1 drastically diminished the number of influenza-infected epithelial cells (Figure 1D). Some PRRSV-1 particles were visible in coincubation conditions (Figure 1D).

We cannot completely rule out that the observed effect of the PRRSV on the replication of swIAV might be related to substances present in the inoculum such as cytokines derived from the infected cells during the amplification process of the virus. However, ultrafiltration and concentration of the viral stocks as well as the dilution of the virus inoculum drastically reduced the concentration of any substances potentially present in the PRRSV-production medium. It is also worth mentioning that PRRSV can inhibit the IFN type 1 protein production and more generally the innate immune response [35]. This feature reduces the chances of having significant concentrations of many cytokines after the amplification of the virus on live cells. On the other hand, the ADV used as a control, is known to induce cytokine production [36], but it did not have any statistically significant impact on the replication of swIAV comparatively to wild-type PRRSV-1. This is additional evidence that the strong reduction of swIAV replication is PRRSV specific.

Moreover, the strain DV of PRRSV-1, an attenuated vaccine strain that is used in PRRSV MLV vaccination protocols, was also capable to affect the replication of swIAV (Figure 1E). Conversely, when PRRSV-1 was replaced by ADV that is able to infect NPTr cells and epithelial cells in general, no reduction of swIAV H1N2 replication was observed (Figure 1E) demonstrating the specificity of the interference we observed. 

### 3.2. The Effect of PRRSV-1 Is till Observed if PRRSV-1 Is Added Shortly before swIAV H1N2 to NPTr Cells

To further analyze the impact of PRRSV-1 exposition on swIAV H1N2 infection, we then carried out PRRSV-1 exposition before (1 to 6 h) or after (1 to 6 h) swIAV H1N2 infection (Figure 2A–C and Appendix A). The more PRRSV-1 was added in advance, the less it interfered with swIAV H1N2 infection (Figure 2B). Indeed, the genomic loads of swIAV H1N2 were significantly more reduced (*p* < 0.01) when PRRSV-1 was added 1 h before swIAV H1N2 compared to the condition when PRRSV-1 was added 6 h before. Similarly, when PRRSV-1 exposition took place after swIAV H1N2 infection, the condition of 1 h delay showed a higher impact on swIAV H1N2 genomic loads compared to the condition with a delay of 6 h (Figure 2C). Regarding the genes involved in type I interferon response, their expression tended to decrease more when PRRSV-1 was added a shorter time in advance (Figure 2B,D). Conversely, their expression increased when PRRSV-1 addition was delayed (Figure 2C,D) (*p* < 0.05). 

### 3.3. Inactivation of Non-Internalized Viruses by Acid Wash Abrogates PRRSV-1 Effect on swIAV H1N2 Infection in NPTr Cells

To further decipher how PRRSV-1 interferes with swIAV H1N2 infection process, we performed acid wash after the incubation of viruses with the cells (Figure 3A). Interestingly, the acid wash which is known to inactivate the non-internalized viruses nearly abrogated the PRRSV-1 effect on swIAV H1N2 infection and on immune gene expression (Figure 3B). Thus, PRRSV-1 impact on swIAV infection occurs mainly from the surface of the epithelial cells and not in intracellular compartments, as expected since PRRSV-1 is described as unable to enter respiratory epithelial cells.

### 3.4. Inactivated PRRSV-1 still Impacts swIAV H1N2 Infection of Tracheal Epithelial Cells

Since PRRSV-1 does not penetrate in epithelial cells, we wanted to determine if PRRSV-1 inhibition of swIAV replication required live viral particles or if it was also possible to observe this inhibition when using inactivated PRRSV-1. As presented in Figure 4 (most significantly affected genes) and Appendix A (all the genes), inactivated PRRSV-1 was still able to interfere with swIAV H1N2 infection of porcine tracheal epithelial cells. However, the inactivated virus seemed less prone than the live virus to alter NPTr-swIAV interactions when added to tracheal epithelial cells simultaneously to swIAV H1N2. Some genes such as MDA5 and IFNβ did not show any significant differences between swIAV H1N2 single infection and coincubation with inactivated PRRSV-1 (Appendix A). On the other hand, viral replication and type I interferon response—from the PRR to the ISG—were always significantly lower when live PRRSV-1 was simultaneously added to swIAV H1N2 infected cells instead of inactivated PRRSV-1 (*p <* 0.01).

### 3.5. PRRSV-1 Impacts swIAV H1N2 Infection in Primary Tissue Lung Slices

Having observed PRRSV-1 interference on swIAV H1N2 infection in epithelial cell culture, we wondered if this effect could be observed in more physiologic settings. We thus proceeded to PRRSV-1 and swIAV H1N2 infections ex vivo on precision-cut lung slices (PCLS) (Figure 5 and Appendix A). Confocal microscopy was used to confirm the host cells of the local virus strains used in this study. We observed actual infection of macrophages (MHC-II-positive alveolar cells) by PRRSV-1 as well as infection of epithelial cells (β-tubulin-positive cells) by swIAV H1N2 (Figure 5B). Swine IAV H1N2 replication detection using RT-qPCR (Figure 5C) confirmed the interference of PRRSV-1 on swIAV H1N2 production in the relevant target tissue. Indeed, swIAV H1N2 replication and type I interferon response were reduced in the presence of live and inactivated PRRSV-1 (Figure 5C), although, as shown with NPTr cells, inactivated PRRSV-1 appeared less effective (except on IFNβ transcript expression where wild-type PRRSV-1 and swIAV H1N2 acted as good inducers unlike inactivated PRRSV-1) and the decrease in the expression was not always significant compared to swIAV H1N2 condition. Thus, in a tissue containing the respective cellular targets of swIAV and PRRSV we made observations similar to what we observed using tracheal epithelial cells that are not susceptible to PRRSV. 

### 3.6. PRRSV-1 and swIAV H1N2-Induced Signaling Pathways in NPTr Cells

We then explored (using western blotting) the main signaling pathways known to be involved in virus infections and host’s antiviral responses (PI3K/AKT, AMPK, MAPK ERK1/2, and JAK/STAT pathways) using NPTr cells. As presented in Figure 6, AKT, AMPK, and JAK2 phosphorylations were decreased in the presence of live PRRSV-1 at 5 and 10 min post-infection (mpi) (Figure 6A,B,D). This significant decrease was observed with PRRSV-1 alone or PRRSV-1 with swIAV H1N2 compared to swIAV H1N2 single infection (*p* < 0.05). In agreement with the previous experiments (Figure 4B), inactivated PRRSV-1 triggered patterns similar to live PRRSV-1, although with a constantly lower impact (Figure 6A,B,D). Conversely, live PRRSV-1, but not inactivated PRRSV-1, triggered ERK phosphorylation, alone or in the presence of swIAV H1N2 (Figure 6C). SwIAV H1N2 had a limited impact on AKT, AMPK, ERK, and JAK2 phosphorylations at 5 and 10 mpi.

## 4. Discussion

In the current study, we were interested in the molecular impact of PRRSV-1 on swIAV H1N2 infection. Indeed, both viruses are frequently encountered in the porcine respiratory tract [4,5,6,7,8] and their interactions are still poorly understood [2]. Some previous studies assessed the consequences of PRRSV/swIAV coinfections regarding virus replications and innate responses of their target cells and tissues [14,15]. However, in these studies, the interactions between PRRSV and tracheal epithelial cells, the main target of swIAV [10] had not been evaluated whereas these cells meet PRRSV viral particles during the infection process [9]. We observed here that PRRSV-1 interacted with tracheal epithelial cells, triggering ERK signaling protein phosphorylation and inhibiting AKT, AMPK, and JAK2 signaling protein phosphorylation. PRRSV-1 was also shown to inhibit swIAV H1N2 infection of epithelial cells when inoculated at the same time or shortly before or shortly after the swIAV, and this is observed, although at a lower level with inactivated PRRSV-1. To our knowledge, infection of porcine epithelial cells in the porcine respiratory tract by PRRSV has never been reported even if PRRSV-2 (but not PRRSV-1), has been shown in vitro to infect some epithelial cells from other species. St-Jude porcine lung (SJPL) cells originally described as an immortalized porcine lung epithelial cell line have been shown to be permissive to PRRSV-2 [25]. However, SJPL cells were actually not porcine cells but monkey cells as evidenced after karyotype and genetic analyses [37]. Two other monkey cell lines, the MARC-145 and CL2621 cells (subclones of MA104 monkey kidney cell line) allow the full replication cycle of PRRSV-2 and are commonly used for PRRSV-2 in vitro propagation. However, porcine tracheal epithelial cells such as NPTr cells were not permissive to PRRSV-1 or PRRSV-2 [17,25]. Thus, different cell surface molecules are needed to allow PRRSV interaction, entry, and infection of cells: porcine sialoadhesin—also known as sialic acid-binding immunoglobulin-type lectin 1 (Siglec-1) or CD169, Siglec-10, CD151, MYH9, and heparan sulfate [38,39,40] have been shown to mediate the interaction of PRRSV with cells; however, only CD163 has been demonstrated as essential for PRRSV genome delivery from endosome to the cytosol of the cells [41] and replication in the porcine host [42].

Regarding swIAV, the hemagglutinin (HA) provides the binding site to interact with the epithelial host cell. More specifically HA binds to a host cell receptor that contains terminal ɑ-2,6-linked or ɑ-2,3-linked sialic acid moieties [10]. Then, the cleavage of HA by cellular proteases is required for fusion and viral infection. NPTr cells are fully susceptible to swIAV and the virus can perform its full cycle in these cells [17,26]. Since PRRSV’s envelope glycoproteins (GPs) such as GP5 contains sialic acid, involved in the interaction with sialoadhesin on macrophages to allow attachment and internalization, we could postulate possible direct interactions between swIAV particles and PRRSV [43,44]. This phenomenon could explain to some extent the inhibition of the swIAV H1N2 replication. This hypothesis is strengthened by the fact that PRRSV’s sialic acid are ɑ-2,3- and ɑ-2,6-linked sialic acids [44]. The probable binding of PRRSV to swIAV particles may lead to the formation of aggregates trapping the swIAV particles and limiting their access to the sialic acids on the cell’s surface. This potential competition between virus sialic acids and cellular sialic acids for swIAV HA would need to be further assessed.

We have observed that PRRSV was able to inhibit AKT, AMPK, and JAK2 phosphorylation and enhance ERK phosphorylation in NPTr cells, however its ligand on these cells is still unknown. We can hypothesize that PRRSV is interacting with heparin sulfates that are ubiquitously expressed on epithelial cells [45,46,47]. Regarding the relation between PRRSV and signaling pathways, it has been shown previously that PRRSV-2 can both activate and inhibit PI3K/AKT pathway in porcine monocyte-derived dendritic cells (Mo-DCs), MARC-145 and AMs depending on the phase of the viral cycle [48,49]. Interestingly, in these studies, heat-inactivated PRRSV-2 failed to inhibit PI3K/AKT in Mo-DCs, indicating that virus replication is essential for this inhibition [48]. Similarly, in our conditions, BPL-inactivated PRRSV-1 failed to inhibit AKT phosphorylation in epithelial cells. Although the inactivation processes were different in these two studies, it can be hypothesized that PRRSV inactivation would still allow the virus to attach to the epithelial cells, trigger receptor-mediated membrane signaling but not replication-dependent signaling. Regarding the AMPK pathway, it has been reported that host cells could antagonize PRRSV-2 infection via the activation of the AMPK-ACC1 signaling pathway [50]. Thus, inhibition of AMPK phosphorylation would help the virus in its multiplication cycle. On the contrary, PRRSV-1 was able to activate the MAPK ERK1/2 pathway in epithelial cells. This observation is in line with a previous observation reporting activation of MAPK ERK1/2 signaling pathway in AMs infected by PRRSV-2 later in the infection process [51]. In that study, UV-inactivated PRRSV-2 was sufficient to trigger ERK phosphorylation, suggesting that the viral entry process may be responsible for early ERK activation [51]. Indeed, UV treatment enables receptor binding and internalization but prevents viral gene synthesis. In our study BPL-inactivated PRRSV-1 was unable to induce ERK phosphorylation suggesting that our inactivated virus was unable to enter epithelial cells while the live virus would be able to enter to some extent, possibly staying in endosomal vesicles. The presence of PRRSV-1 particles in some epithelial cells suggested in our study by confocal microscopy would support this hypothesis as well as previous studies reporting the presence of PRRSV antigen in non-macrophage cells [14,52,53]. On the contrary, the results obtained following the acid wash applied after incubation of the cells with the viruses suggest that most of the PRRSV-mediated interference is linked to non-internalized viral particles.

SwIAV needs to activate MAPK ERK1/2 and PI3K signaling pathways to mediate the vacuolar (H^+^)-ATPases-dependent intracellular pH change that is required for endosome fusion [54]. We observed induction of this pathway in NPTr cells 60 min post-swIAV-infection and later (data not shown) indicating a reaction of the cells to the swIAV H1N2 infection process. The decreased activation of PI3K/AKT could partially explain the inefficient swIAV H1N2 infection of NPTr when PRRSV-1 was co-applied to the cells. However, AKT phosphorylation inhibition was observed with live PRRSV-1 but not inactivated PRRSV-1, whereas inactivated PRRSV-1 was still able to inhibit swIAV replication as well, indicating that other mechanisms are involved. Indeed, the only signaling pathway commonly impacted by live and inactivated PRRSV was AMPK. We can thus hypothesize that PRRSV-1 ligand on epithelial cells might trigger an AMPK inhibitory signal. Interestingly, it has been recently reported that AMPK mediated autophagy promoted IAV replication [55], thus PRRSV-mediated AMPK inhibition could be responsible for the decrease in swIAV replication. 

As per the JAK-STAT signaling pathways, it plays an important role in swIAV infection. Following IFNα/β production and binding to their receptors (IFNAR), JAK-STAT signaling leads to the recruitment and phosphorylation of IRF9 into the STAT1/STAT2 heterodimer to make ISG factor 3 (ISGF3) [56]. Translocation of ISGF3 to the nucleus induces the transcription of the different ISGs contributing to the antiviral defense against swIAV infection [57]. On the other hand, PRRSV has been described to inhibit the IFN-activated JAK-STAT signal transduction and ISG expression in MARC-l45 and AM cells [58]. In order to evade the host antiviral response, PRRSV is capable of inhibiting IFN-activated JAK-STAT signaling by blocking the ISGF3 nuclear translocation [59]. As a consequence, studies showed lower transcript levels of ISG15 and ISG56 and lower protein levels of STAT2 in PRRSV infected cells following IFN stimulation [60]. Our findings confirmed that PRRSV-1 interferes with the JAK-STAT signaling in NPTr cells, by downregulating the phosphorylation of JAK2 and leading to the low expression of ISGs even after swIAV infection.

Overall, PRRSV may interfere with swIAV early infection over two levels, (i) direct inhibition of swIAV infection/replication of epithelial cells; (ii) modification of the host’s innate immune response, mainly through AM infection and killing [9]. Herein we have shown that coinoculation with PRRSV-1 might impact swIAV H1N2 capacity of replication in a respiratory epithelial cell line and lung tissue. Conversely, the inhibition of innate immune response by PRRSV and especially its capacity to antagonize antiviral interferon response using its nonstructural proteins [58] would facilitate swIAV replication. We thus propose that the balance between these phenomena might explain the discrepancies observed between different in vivo coinfection experiments, for which PRRSV primary infection increases, decreases or has no impact on swIAV co/superinfections. Finally, it is also known that swIAV infects unproductively alveolar macrophages (for review see [61]), the main target of PRRSV, which raises the possibility of a reverse interference of swIAV on PRRSV replication, adding another level of complexity in the interactions between these two viruses. 

## 5. Conclusions

In conclusion, our study is showing for the first time that PRRSV can alter the relation between swIAV and its main target cells opening the doors to further studies on the interplay between respiratory viruses. Interference between PRRSV and swIAV could also have consequences regarding vaccination, as PRRSV vaccination involves MLV that showed a similar impact as a wild-type live virus. Based on our results, further investigations would be necessary to evaluate how PRRSV vaccination could modulate a porcine host’s susceptibility to concomitant swIAV infection, potentially offering indirect and short term heterologous protection to the pigs.

## Figures and Tables

**Figure 1 vaccines-08-00508-f001:**
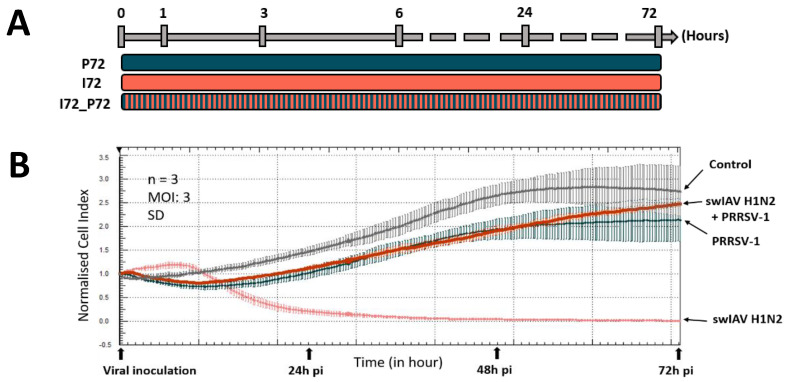
Protocol (**A**), normalized cell index (**B**), relative viral loads and relative expression of cellular transcripts (**C** and **E**), and immunostaining (**D**) of the different conditions in newborn pig tracheal cells showing the negative impact of PRRSV-1 on swIAV H1N2 replication and NPTr cells response. Cells were infected at a MOI of 3, *n*= 6 wells per condition for qPCR and *n*=3 for the impedancemetry. Mean values and standard deviations are represented. C0 and C24 stand for the non-infected conditions over 0 h and 24 h respectively, P24 and P72 stand for PRRSV-1 stimulation over 24 h, and 72 h, I24, and I72 for swIAV H1N2 infection over 24 h and 72 h, A24 for ADV infection over 24 h, MLV24 for stimulation with the DV strain of PRRSV-1 over 24 h, I24_P24, I24_A24, I24_MLV24, and I72_P72 for the coincubation of 2 viruses over 24 h or 72 h. Comparisons were carried out using a one-way ANOVA test and Tukey’s post-test. Differences were considered significant when *p* < 0.05 (*) or *p* < 0.01 (**). NPTr cells (**D**) were fixed after 24 h and stained with an antibody against viral N protein to detect PRRSV-1 particles and PRRSV-1 infected cells (in green—white arrows) and an anti-nucleoprotein polyclonal antibody to detect swIAV H1N2 particles and infected cells (in red). Cell nuclei were stained (in white) using 4′,6′-diamidino-2-phenylindole (DAPI). Images were generated using a laser-scanning confocal microscope.

**Figure 2 vaccines-08-00508-f002:**
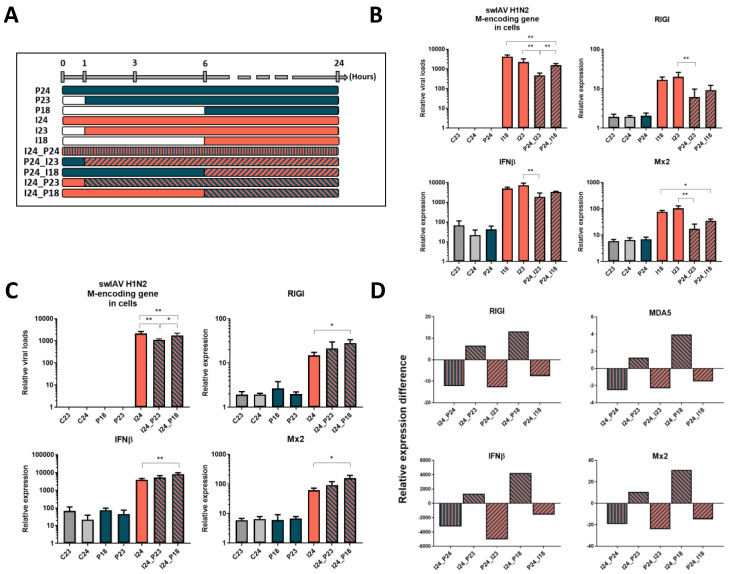
Protocol (**A**) and relative expression of viral and cellular transcripts (**B**,**C**) of the different conditions in newborn pig tracheal cells showing the impact of the delay and the order of the viruses on swIAV H1N2 replication and NPTr cells response. Difference of the relative expression between dual stimulations and influenza single infections is represented in panel (**D**). Cells were infected at a MOI of 3, *n*= 6 wells per condition for qPCR. C23 and C24 stand for the non-infected conditions over 23 h and 24 h respectively, P24 stands for PRRSV-1 stimulation over 24 h, I24 for swIAV H1N2 infection over 24 h, I24_P24, P24_I23, P24_I18 stand for dual administrations (PRRSV-1 then swIAV) of both viruses with a delay of 0, 1, and 6 h respectively. I24_P23 and I24_P18 stand for dual administration after reversing the viruses (swIAV then PRRSV-1) order with a delay of 1 and 6 h respectively. Mean values and standard deviations are represented. Comparisons were carried out using a one-way ANOVA test and Tukey’s post-test after sorting the data by rank. Differences were considered significant when *p* < 0.05 (*) or *p* < 0.01 (**).

**Figure 3 vaccines-08-00508-f003:**
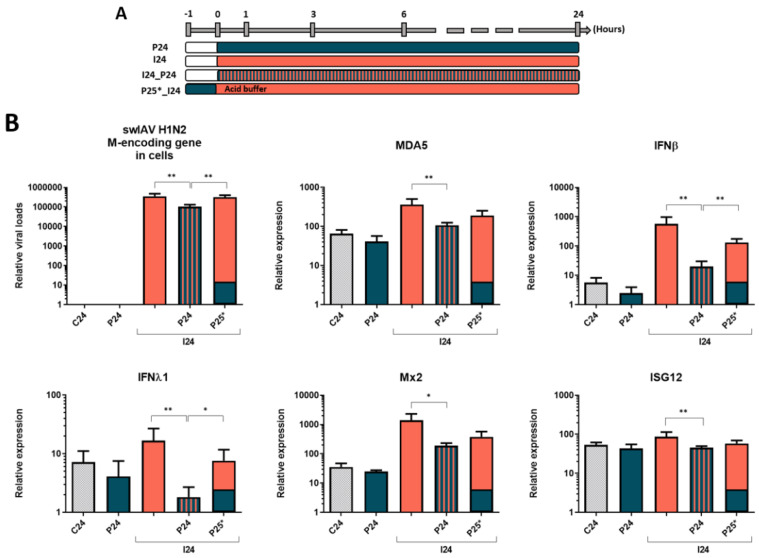
Protocol (**A**) and relative viral loads and relative expression of cellular transcripts (**B**) showing the impact of the inactivation of non-penetrated PRRSV-1 particles by the acid buffer wash on swIAV H1N2 replication and NPTr cells response. Cells were infected at an MOI of 3, *n*= 6 wells per condition for qPCR. C24 stands for the non-infected condition over 24 h, P24 stands for PRRSV-1 stimulation over 24 h, I24 for swIAV H1N2 infection over 24 h, I24_P24 for the coincubation of both viruses over 24 h, and P25*_I24 for the stimulation with PRRSV-1 for 1 h then infection with swIAV H1N2 for 24 h after an acid buffer wash. Mean values and standard deviations are represented. Comparisons were carried out using a one-way ANOVA test and Tukey’s post-test. Differences were considered significant when *p* < 0.05 (*) or *p* < 0.01 (**).

**Figure 4 vaccines-08-00508-f004:**
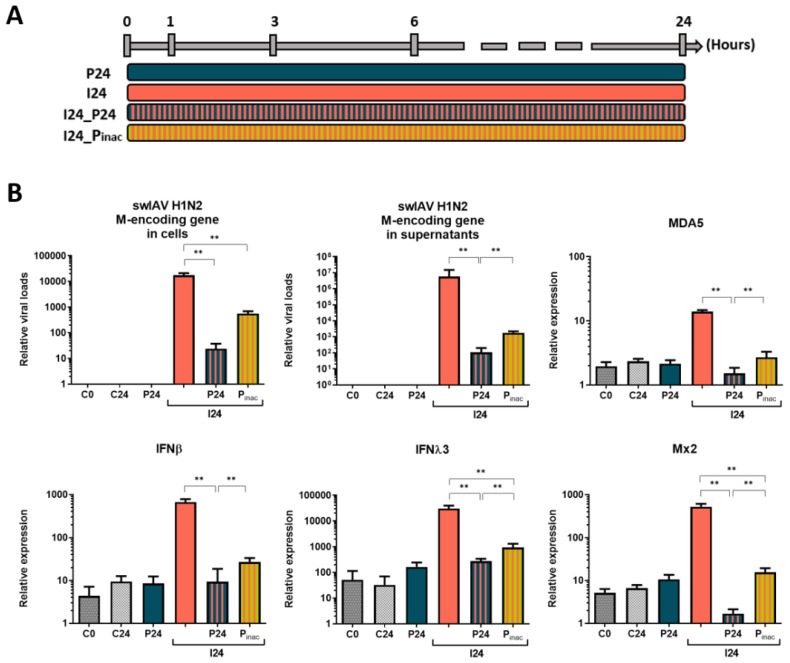
Protocol (**A**) and relative viral loads and relative expression of cellular transcripts (**B**) of the different conditions in newborn pig tracheal cells showing the impact of inactivated PRRSV-1 on swIAV replication and NPTr cells response compared to live PRRSV-1. Cells were infected at an MOI of 3, *n*= 6 wells per condition for qPCR. C0 and C24 stand for the non-infected conditions over 0 h and 24 h respectively, P24 stands for PRRSV-1 stimulation over 24 h, I24 for swIAV H1N2 infection over 24 h, I24_P24 and I24_Pinac for the coincubation of both viruses over 24 h using live and inactivated PRRSV-1, respectively. Mean values and standard deviations are represented. Comparisons were carried out using a one-way ANOVA test and Tukey’s post-test. Differences were considered significant when *p* < 0.05 (*) or *p* < 0.01 (**).

**Figure 5 vaccines-08-00508-f005:**
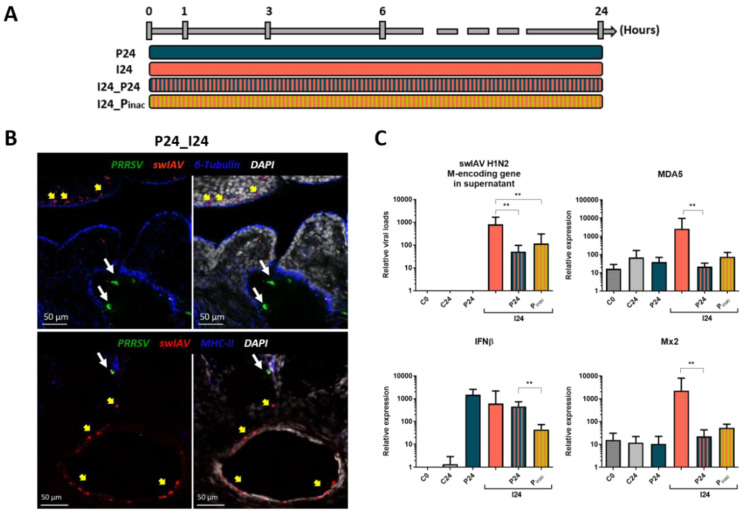
Protocol (**A**), immunostaining (**B**), and relative viral loads and relative expression of cellular transcripts (**C**) of the different conditions in precision-cut lung slices (PCLS) showing the impact of live and inactivated PRRSV-1 on swIAV replication and cell response. PCLS were coinfected with 10^5^ particles of swIAV H1N2 and/or 10^5^ particles of live or inactivated PRRSV-1. After 24 h, PCLS were fixed and stained with an antibody against viral N protein to detect PRRSV-1 particles (in green) and PRRSV-1 infected cells (white arrows) and an anti-nucleoprotein polyclonal antibody to detect swIAV H1N2 (in red) and infected cells (yellow arrows). Anti-beta-tubulin monoclonal antibody was used for epithelial ciliated cells staining while anti-MHC-II antibody was used for macrophages and other MHC-II presenting cells staining (in blue). Cell nuclei were stained (in white) using 4′,6′-diamidino-2-phenylindole (DAPI). Images were generated using a laser-scanning confocal microscope. *n*= 10 slices per condition for qPCR. C0 and C24 stand for the non-infected conditions over 0 h and 24 h respectively, P24 stands for PRRSV-1 infection over 24 h, I24 for swIAV H1N2 infection over 24 h, I24_P24 and I24_Pinac for the coinfection with both viruses over 24 h using active and inactivated PRRSV-1 respectively. Mean values and standard deviations are represented. Comparisons were carried out using a one-way ANOVA test and Tukey’s post-test after sorting the data by rank. Differences were considered significant when *p* < 0.05 (*) or *p* < 0.01 (**).

**Figure 6 vaccines-08-00508-f006:**
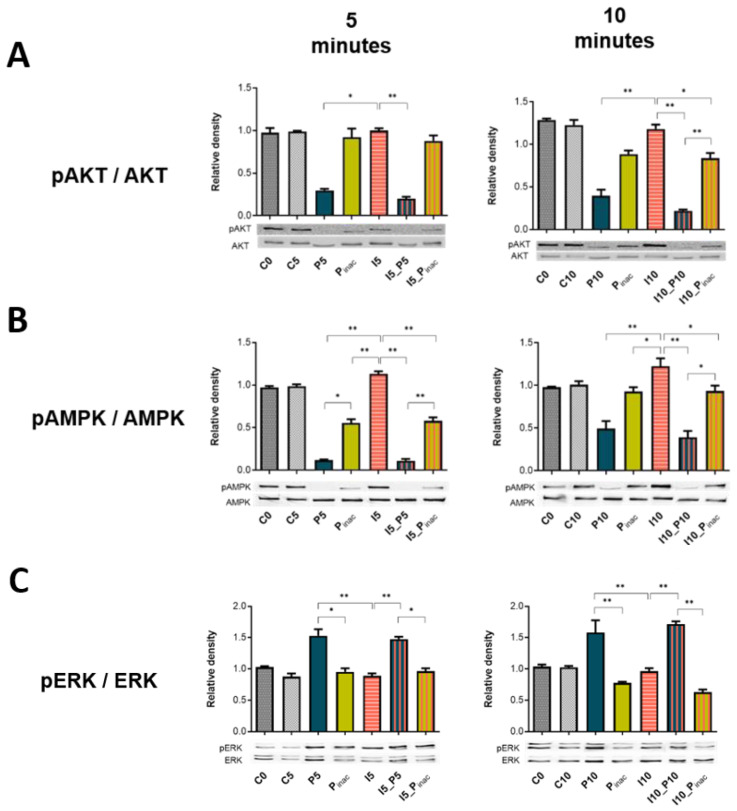
Western blots of phospho-AKT (**A**), phospho-AMPK (**B**), phospho-ERK1/2 (**C**), and phospho-JAK2 (**D**) in newborn pig tracheal cells infected with swIAV H1N2 in the absence or presence of PRRSV-1. Cells were infected at an MOI of 3 and harvested simultaneously at 5 and 10 min, so the results are representative of two independent experiments. C0, C5, and C10 stand for the cells from the non-infected conditions over 0, 5, and 10 min; P5, P10 and Pinac for cells stimulated with active or inactivated PRRSV-1 simultaneously for 5 and 10 min, while I5 and I10 stand for cells infected with swIAV H1N2 for 5 or 10 min. AKT, AMPK, ERK, and JAK2 are shown as loading controls and did not change with each condition over time. Mean values and standard deviations are represented, *n*= 3 wells per condition. Comparisons were carried out using a one-way ANOVA test and Tukey’s post-test after sorting the data by rank. Differences were considered significant when *p* < 0.05 (*) or *p* < 0.01 (**).

**Table 1 vaccines-08-00508-t001:** List of the primers used in the study.

Primer Abbreviation and Full Name	Primer Sequences: Sense (S) and Anti-Sense (AS)	Amplicon Size (bp)	Annealing Temperature (°C)	Accession Number or PMIDs
Viruses				
SwIAV H1N2 M-encoding gene	(S) AGATGAGTCTTCTAACCGAGGTCG (AS) TGCAAAAACATCTTCAAGTCTCTG (P) (6FAM)-TCAGGCCCCCTCAAAGCCGA-(TAM)	100	60	15460317
PRRSV *Finistère* ORF5	(S) AGAACCAGCGCCAATTCAGA (AS) TCTTTTTCGCCTGTCCTCCC (P) (HEX)-AAACACAGCTCCAATGGGGAATGGC-(TAM)	135	60	28241868
ADV gB-encoding gene	(S) GCGGGTACGTGTACTACGAG (AS) GAGGCCCTGGAAGAAGTTGG (P) (6FAM)-ACTACAGCTACGTGCGCATGGTGGAG-(TAM)	287	63	NC_006151
Reference genes				
B2MI *Beta-2-microgobulin*	(S) CAAGATAGTTAAGTGGGATCGAGAC (AS) TGGTAACATCAATACGATTTCTGA	161	58	17697375
HPRT1 *Hypoxanthine phosphoribosyltransferase 1*	(S) GGACTTGAATCATGTTTGTG (AS) CAGATGTTTCCAAACTCAAC	91	60	17697375
GAPDH *Glyceraldehyde 3-phosphate dehydrogenase*	(S) CTTCACGACCATGGAGAAGG (AS) CCAAGCAGTTGGTGGTACAG	170	63	AF017079
Pattern Recognition Receptors				
MDA5 *Melanoma differentiation-associated protein 5*	(S) AGCCCACCATCTGATTGGAG (AS) TTCTTCTGCCACCGTGGTAG	133	60	MF358967.1
RIGI *Retinoic acid-inducible gene I*	(S) CGACATTGCTCAGTGCAATC (AS) TCAGCGTTAGCAGTCAGAAG	126	60	NM_213804
TLR2 *Toll-like receptor 2*	(S) ACGGACTGTGGTGCATGAAG (AS) GGACACGAAAGCGTCATAGC	101	62	NM_213761.1
TLR3 *Toll-like receptor 3*	(S) GACCTCCCGGCAAATATAAC (AS) GGGAGACTTTGGCACAATTC	155	60	NM_001097444
TLR4 *Toll-like receptor 4*	(S) TGTGCGTGTGAACACCAGAC (AS) AGGTGGCGTTCCTGAAACTC	136	62	NM_001293316.1
TLR6 *Toll-like receptor 6*	(S) TCCCAGGATCAAGGTTCTTG (AS) GAGCAGAGTCCCCTTATAAC	370	60	NM_213760.2
TLR7 *Toll-like receptor 7*	(S) CGGTGTTTGTGATGACAGAC (AS) AACTCCCACAGAGCCTCTTC	174	62	NM_001097434.1
TLR8 *Toll-like receptor 8*	(S) CACATTTGCCCGGTATCAAG (AS) TGTGTCACTCCTGCTATTCG	145	58	NM_214187.1
TLR9 *Toll-like receptor 9*	(S) GGCCTTCAGCTTCACCTTGG (AS) GGTCAGCGGCACAAACTGAG	151	64	NM_213958.1
TLR10 *Toll-like receptor 10*	(S) CTTTGATCTGCCCTGGTATCTCA (AS) CATGTCCGTGCCCACTGAC	51	60	AB_208699,1
Interferons				
IFNα *Interferon alpha*	(S) GGCTCTGGTGCATGAGATGC (AS) CAGCCAGGATGGAGTCCTCC	197	62	JQ839262.1
IFNβ *Interferon beta*	(S) AGTTGCCTGGGACTCCTCAA (AS) CCTCAGGGACCTCAAAGTTCAT	70	60	21645029
IFNλ1 *Swine interferon lamba 1*	(S) GAGGCTGAGCTAGACTTGAC (AS) CCTGAAGTTCGACGTGGATG	115	60	NM_001142837
IFNλ3 *Swine interferon lamba 3*	(S) CCTGGAAGCCTCTGTCATGT (AS) TCTCCACTGGCGACACATT	72	60	29677213
Interferon-stimulated genes				
PKR *Protein kinase R*	(S) CACATCGGCTTCAGAGTCAG (AS) GGGCGAGGTAAATGTAGGTG	166	61	NM_214319
OAS1 *2’-5’-Oligoadenylate Synthetase 1*	(S) CCCTGTTCGCGTCTCCAAAG (AS) GCGGGCAGGACATCAAACTC	303	64	NM_214303
MX1 *Myxovirus resistance protein 1*	(S) AGTGTCGGCTGTTTACCAAG (AS) TTCACAAACCCTGGCAACTC	151	60	NM_214061
MX2 *Myxovirus resistance protein 2*	(S) CCGACTTCAGTTCAGGATGG (AS) ACAGGAGACGGTCCGTTTAC	156	62	AB258432
ISG12 *Interferon-stimulated gene 12*	(S) GTACTTCCTCCCTGATAGG (AS) ACAGCTACAGAAGCCTTG	76	54	NM_001198921.1
ISG15 *Interferon stimulated gene 15*	(S) GATCGGTGTGCCTGCCTTC (AS) CGTTGCTGCGACCCTTGT	176	58	NM_001128469.3

**Table 2 vaccines-08-00508-t002:** Antibodies used for western blotting.

Targeted Protein	Specific Antibody
-Phospho-AKT	-Rabbit polyclonal anti-phospho-AKT (Ser473) #9271 (Ozyme)
-AKT	-Rabbit monoclonal anti-AKT (11E7) #4685 (Ozyme)
-Phospho-AMPK -AMPK	-Rabbit monoclonal anti-phospho-AMPK alpha (Thr172) (40H9) #2535L (Ozyme) -Rabbit polyclonal anti-Total AMPK #2532L (Ozyme)
-Phospho-ERK1/2 -ERK2	-Rabbit monoclonal anti-phospho-p44/42 MAPK (Erk1/2) (Thr202/Tyr204) (D13.14.4E) #4370 (Ozyme) -Rabbit polyclonal anti-ERK2 (GTX27948) (Tebu-bio)
-Phospho-JAK2 -JAK2	-Rabbit polyclonal anti-phospho-JAK2 (Tyr1007/1008) #3771 (Ozyme) -Rabbit monoclonal anti-JAK2 (D2E12) #3230 (Ozyme)

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
