# Peer review of "Porcine Reproductive and Respiratory Syndrome Virus Interferes with Swine Influenza A Virus Infection of Epithelial Cells"

_vaccines, 2020, doi:10.3390/vaccines8030508_

Round 1

Reviewer 1 Report

In this study, the authors explore in detail the previously reported interference effects between PRRSV and swine influenza A virus, in an attempt to elucidate the mechanism.   All in all, it is a nicely done study on a difficult experimental system.  I think the reported effects are likely real, and interesting.  My main concerns are for the interpretation of the data, which seems to me to omit an important alternate explanation. 

Major points:

  1. I have a concern that the PRRSV is in cell culture medium derived from infected cells, which would contain a range of signaling molecules in addition to the virus, and that the observed effects come from a component of the inoculum other than the virus. Please either demonstrate that the virus itself, in a purified state independent of its cell culture medium, is driving the effect, or modify the conclusions and abstract to include this alternate explanation. 

I think the experiments in Figs. 3 and 4 seem to have been adequately carried out, but the reduction rather than abrogation of the interference effect with heat inactivation and acid wash suggest to me that the observed effect may be due to some component other than PRRSV in the inoculum. 

  1. I don’t see a reference to back up the effectiveness of acid inactivation of non-internalized virus in the cited reference (listed in the methods as reference 22). Please check this and all references for errors.

  1. Ethical concern - Animal welfare doesn’t seem to be mentioned in detail that I can see, though much of the material seems to be primary animal-derived. This isn’t a major concern in terms of the results, but needs to be present in a study like this.  I don't necessarily see anything other than the absence of this section to cause concern, but I can't review what isn't there.

Author Response

Reviewer 1 comments: In this study, the authors explore in detail the previously reported interference effects between PRRSV and swine influenza A virus, in an attempt to elucidate the mechanism. All in all, it is a nicely done study on a difficult experimental system. I think the reported effects are likely real, and interesting. My main concerns are for the interpretation of the data, which seems to me to omit an important alternate explanation.

Thanks for the very positive comment. For modifications see the attached "report notes" file (modifications in yellow in the text)

Major points

  1. I have a concern that the PRRSV is in cell culture medium derived from infected cells, which would contain a range of signaling molecules in addition to the virus, and that the observed effects come from a component of the inoculum other than the virus. Please either demonstrate that the virus itself, in a purified state independent of its cell culture medium, is driving the effect, or modify the conclusions and abstract to include this alternate explanation.

I think the experiments in Figs 3 and 4 seem to have been adequately carried out, but the reduction rather than abrogation of the interference effect with heat inactivation and acid wash suggest to me that the observed effect may be due to some component other than PRRSV in the inoculum.

Before inoculation, ultrafiltration of the viruses was carried out on using Amicon® Ultra-15 Centrifugal Filter Unit (reference number UFC910024 – pore size 100 kDa Nominal Molecular Weight cut-off). Thus, the presence of cytokines or any other signaling molecules (going through the filter) should be massively reduced at this stage. During this process, a 10-fold reduction of the volume of the viral inoculum is reached. For the inoculation step, we consequently used very low volumes of inoculum and carried out a second dilution (20 to 50 µl of virus in a total volume of 200 µl) which minimalized the effect of any signaling effective substances in the PRRSV medium.

It is also important to mention that PRRSV is a good inhibitor of the innate immune response especially IFN type 1 production, which reduces the chances of having significant concentrations of other cytokines/chemokines after the amplification of the virus on live cells.

On the other hand, the ADV used as a control, is known to induce cytokine production, however, it did not have any statsically significant effect on the replication of swIAV. This is an additional evidence that the inhibition of swIAV replication is specific and related to PRRSV particles.

However, this possibility cannot be totally ruled out, and consequently we have added some sentences about it in the manuscript (see lines 116 and 336-346).

  1. I do not see a reference to back up the effectiveness of acid inactivation of non-internalized virus in the cited reference (listed in the methods as reference 22). Please check this and all references for errors.

The acid wash was used in the penetration and growth kinetic assays in different papers (Meurens et al. 2004; Song et al. 2019). The reference detailing the technique (Mettenleiter 1989) has been added to the text (see lines 177-178).

  1. Ethical concern - Animal welfare doesn’t seem to be mentioned in detail that I can see, though much of the material seems to be primary animal-derived. This isn’t a major concern in terms of the results, but needs to be present in a study like this. I don't necessarily see anything other than the absence of this section to cause concern, but I can't review what isn't there.

That is right. We thank you for the comment. However, to reduce the use of experimental animals the lungs were collected from pigs slaughtered in the course of the regular management of the herds. As a consequence, no trial number has been attributed since the experimental authorization was not necessary in this specific case. This point has been specified in the manuscript (see lines 84 and 86-89).

Reviewer 2 Report

The manuscript describes the interference of PRRSV-1 on swIAV infection in its target cell line-epithelial cells. It is well organized. However, the data provided in some of the figures and tables are not match. Some of the data analysis also needs to be redone. Here are some comments:

Major concerns:

  1. Fig 1 showed results from cells infected with swIAV H1N2 after PRRSV-1 stimulation, while Table 3 displayed results from cells infected with swIAV H1N2 first and then PRRSV-1 stimulation. It is not consistent. Also, table 3 did not provide correct information to Fig 2. Did the authors provide the wrong table? Please check.
  2. In Fig 1E, why the authors presented swIAV H1N2 viral load from MLV24_I24 group in supernatants, but swIAV H1N2 viral load from A24_I24 in cells? Is it better to use the data both in supernatants or cells?
  3. In Fig 4, what is the order of the co-infection? According to line 397 “P24_I24 and Pinac_I24”, did the authors infect cells with PRRSV-1 or inactivated PRRSV-1 first and then swIAV? If yes, the data in Fig 4 is not consistent with the data in Table 4 because the infection order in table 4 is swIAV first and then PRRSV-1. Please clarify.
  4. In Table 4, the statistical analysis for MDA5 and IFNβ between I24 and I24_Pinac as well as I24_P24 and I24_Pinac does not make sense. Same in the fig 4. Please clarify.
  5. Line 386-387, description about the single infection and coincubation is wrong. They have significant difference according to the data in Fig 4 and Table 4. Please check.
  6. In Fig 5B, what is the purpose of fig 5B? If the authors want to show the presence of PRRSV-1 particles in the epithelial cells, please emphasize it in the results part.
  7. In Fig 5c, check the statistic analysis about MDA5, IFNβ and Mx2. Also, IFNβ decreased more in Pinac-I24 group than that in P24-I24 group. Please explain why.
  8. According to authors’ description in line 405, Table 5 should be the data about P24_I24 and Pinac_I24 treatments. But table 5 displayed I24_P24 and I24_Pinac treatments. Did the authors provide the wrong table?
  9. All the tables are better provided as supplementary material.

Minor concerns:

  1. Line 170, change “H1N2” to “H1N2,”
  2. Line 345, add “when” to the front of “the more”
  3. Fig 2C, to this reviewer, the expression level of swIAV H1N2 M-encoding gene in cells between I24 and I24-P18 groups has no difference. But the authors labeled ** in the figure. Please clarify it.
  4. Line 408, RT-qPCR results showed in Fig 5C, not in 5B
  5. Line 497-498, unclear description, what does “The binding of PRRSV to swIAV” mean?
  6. Line 565, change “than” to “as”

Author Response

Reviewer 2 comments: The manuscript describes the interference of PRRSV-1 on swIAV infection in its target cell line-epithelial cells. It is well organized. However, the data provided in some of the figures and tables are not match. Some of the data analysis also needs to be redone. Here are some comments:

Thanks for the careful and useful revision of the manuscript. For modifications see also attached file - report notes file (modifications in yellow in the text)

Major concerns

  1. Fig 1 showed results from cells infected with swIAV H1N2 after PRRSV-1 stimulation, while Table 3 displayed results from cells infected with swIAV H1N2 first and then PRRSV-1 stimulation. It is not consistent. Also, table 3 did not provide correct information to Fig 2. Did the authors provide the wrong table? Please check.

Thanks for this comment. In Figure 1, both viruses were used together at the same time (coincubation). Indeed, we unintentionally used 2 abbreviations for the same conditions, there is no difference between I24_P24 and P24_I24. Both viruses were in contact with the cells during 24 h. To clarify this point, the figures has been modified to match the tables and we used I24_P24 everywhere (see Figure 1).

Table 3 (now supplementary Table 1) has been modified to represent only the results showed in Figure 1 and a new table has been added (see supplementary Table 2) to provide more details and data regarding the experiments presented on Figure 2.

  1. In Fig 1E, why the authors presented swIAV H1N2 viral load from MLV24_I24 group in supernatants, but swIAV H1N2 viral load from A24_I24 in cells? Is it better to use the data both in supernatants or cells?

Thanks for the comment. Following reviewer 2 comment, the measure of the viral loads in the supernatant coming from the experiment including ADV was done and was added to the figure as suggested (see Figure 1).

  1. In Fig 4, what is the order of the co-infection? According to line 397 “P24_I24 and Pinac_I24”, did the authors infect cells with PRRSV-1 or inactivated PRRSV-1 first and then swIAV? If yes, the data in Fig 4 is not consistent with the data in Table 4 because the infection order in table 4 is swIAV first and then PRRSV-1. Please clarify.

There is no time delay for coincubation conditions, when we talk about coincubation, it means that both viruses were added at the same time. In Figure 4, we did not change the delay between the viruses, which explains why both viruses remained present for 24 h.

  1. In Table 4, the statistical analysis for MDA5 and IFNβ between I24 and I24_Pinac as well as I24_P24 and I24_Pinac does not make sense. Same in the fig 4. Please clarify.

Thanks for this comment. Relative expressions of MDA5 and IFNβ transcripts between I24 and I24_Pinac look different once represented on the graph since we used a logarithmic scale that could be visually misleading. However, these differences are not statically significant (P-values were >0.05). For MDA5, P = 0.0512 and for IFNβ, P = 0.1479.

  1. Line 386-387, description about the single infection and coincubation is wrong. They have significant difference according to the data in Fig 4 and Table 4. Please check.

As explained before, the results were not significantly different.

  1. In Fig 5B, what is the purpose of fig 5B? If the authors want to show the presence of PRRSV-1 particles in the epithelial cells, please emphasize it in the results part.

Thanks. The purpose of the fig 5B is to confirm the host cells of the viruses since the local strains we are using were never evaluated on pulmonary tissue using confocal microscopy (added in the results section, see lines 422-424).

  1. In Fig 5c, check the statistical analysis about MDA5, IFNβ and Mx2. Also, IFNβ decreased more in Pinac-I24 group than that in P24-I24 group. Please explain why.

The statistical analyses were reviewed and the presented results appear correct. In MDA5 and Mx2, we can notice a slight decrease in Pinac_I24 compared to I24, however this difference was not statistically significant (P = 0.1606 in MDA5 and P = 0.2512 in Mx2 for I24 vs Pina_I24).

As per the decrease in the condition Pinac_I24 for IFNβ. It is due to the absence of the effect of activated PRRSV that is an inducer of IFN type 1 transcripts (while inhibiting post-transcriptional IFN production). If we compare P24 with the control condition, we can see that PRRSV induces IFNβ transcripts expression in PCLS, it is also the case when mixing it with swIAV. However, once inactivated, PRRSV was not able to induce IFNβ transcripts to the same extent anymore (not shown). This could explain this significant decrease in the coincubation condition with the inactivated PRRSV. We believe that the remaining expression is due to swIAV in this condition but this effect was reduced since the inactivated particles of PRRSV can still interact with swIAV and reduce its adhesion to the cells (text slightly modified lines 429-431).

  1. According to authors’ description in line 405, Table 5 should be the data about P24_I24 and Pinac_I24 treatments. But table 5 displayed I24_P24 and I24_Pinac treatments. Did the authors provide the wrong table?

Thanks for inquiring. There is no difference between I24_P24 and P24_I24. It is a coincubation condition which means that both viruses were inoculated at the same time. Both viruses were in contact with the cells during 24 h (the figures has been modified to match the tables and we used I24_P24 everywhere).

  1. All the tables are better provided as supplementary material.

We agree, thanks, and all tables 3-5 are now provided as supplementary material. However, we kept tables 1 and 2 which can help to better assess mat and meth.

Minor concerns

  1. Line 170, change “H1N2” to “H1N2,”

Thanks. Done (see line 175)

  1. Line 345, add “when” to the front of “the more”

Done as suggested (see line 363)

  1. Fig 2C, to this reviewer, the expression level of swIAV H1N2 M-encoding gene in cells between I24 and I24-P18 groups has no difference. But the authors labeled ** in the figure. Please clarify it.

The statistical analysis has been double checked and there is a significant difference between I24 and I24_P18 (P value = 0.0083). The figures might be visually misleading since we are using a logarithmic scale and the scale is not the same between the figures of the different transcripts.

  1. Line 408, RT-qPCR results showed in Fig 5C, not in 5B

Corrected in the text (see line 426)

  1. Line 497-498, unclear description, what does “The binding of PRRSV to swIAV” mean?

We are hypothesizing a possible adhesion between swIAV and PRRSV particles, since PRRSV expresses sialic acids on its surface. These sialic acids are the receptors of swIAV on epithelial cells. Such interaction may lead to the formation of aggregates trapping the swIAV particles and preventing their adhesion to the cell’s surface. For clarity, the sentence has been modified (see lines 514-516).

  1. Line 565, change “than” to “as”

Thanks. It has been modified (see line 584).

Round 2

Reviewer 1 Report

Thanks for addressing my concerns, which I think has strengthened the manuscript.  

Author Response

Dear Reviewer, Dear Editor,

We would like to thank you for the interest you have expressed to our article and for the constructive remarks as well.

We have modified the revised manuscript to answer the last criticisms and comments. Please find below a point per point answer to the raised concerns (in yellow in the manuscript).

English has been further revised in the paper with small modifications (see lines 27-29, 95-96, 117, 313, 345-346, 354-355, 379, 384, 400, 423, and 485).

Reviewer 2 comments: Several minor concerns about this revised version:

  1. The supplementary material file does not have tables, only figures. Please check it.

Thanks and sorry. Hopefuly, the tables are provided at the right place in the website now. I also put it below to make sure you see it (however, data are correct but table’s aspects have been slightly modified by the copy/paste process here).

  1. Since the authors decide to use I24_P24 for coincubation, please change all the P24_I24 in the figure legends to I24_P24.

Thanks, it has been modified (see lines 312, 313, 376, 397, 421, 457).

  1. In table 4, for MDA5, I24_P24 (1.51±0.3) vs I24_Pinac (2.69±0.5) is significantly different (p<0.0001), but I24 (13.87±0.8) vs I24_Pinac (2.69±0.5) is not (ns). Please explain why? Same question for IFNβ. I24_P24 (9.29±9.3) vs I24_Pinac (26.78±6.5) is significantly different (p<0.0001), but I24 (665.89±116.5) vs I24_Pinac (26.78±6.5) is not (ns). (Since the reviewer could not find the tables in supplementary file, this question still uses data in table 4 in the older version).

This is related to the ranking process (see statistical analysis lines 279 to 281). The large difference between two values of relative expression is reduced to a difference of one rank. Despite the difference between these two values, if there are no other values in between, the attributed ranks are consecutive and the difference is only of one rank. This process is responsible of a reduction in the difference the condition I24 and I24_Pinac. This can also happen with some non-parametric tests (such as Kruskall-Wallis for instance).

The ns p-values were p = 0.0512 for MDA5 and p = 0.1479 for IFNß.

  1. In table 5, for IFNβ, I24_P24 (457.14±274.2) vs I24_Pinac (44.01±28.6) is significantly different (p=0.0027), but I24 (615.66±1558.2) vs I24_Pinac (44.01±28.6) is not (ns). Why? Large standard deviation?

Yes, this is part of the explanation and see also above. The p-value was p = 0.4427

Reviewer 2 Report

Several minor concerns about this revised version:

  1. The supplementary material file does not have tables, only figures. Please check it.
  2. Since the authors decide to use I24_P24 for coincubation, please change all the P24_I24 in the figure legends to I24_P24
  3. In table 4, for MDA5, I24_P24 (1.51±3) vs I24_Pinac (2.69±0.5) is significantly different (p<0.0001), but I24 (13.87±0.2) vs I24_Pinac (2.69±0.5) is not (ns). Please explain why? Same question for IFNβ. I24_P24 (9.23±0.9.3) vs I24_Pinac (26.78±6.5) is significantly different (p<0.0001), but I24 (665.89±116.5) vs I24_Pinac (26.78±6.5) is not (ns). (Since the reviewer could not find the tables in supplementary file, this question still uses data in table 4 in the older version)
  4. In table 5, for IFNβ, I24_P24 (457.14±2) vs I24_Pinac (44.01±28.6) is significantly different (p=0.0027), but I24 (615.66±1558.2) vs I24_Pinac ((44.01±28.6) is not (ns). Why? Large standard deviation?

Author Response

(The authors gave the same response as above.)
